# The Citrus Flavonoid Hesperetin Has an Inadequate Anti-Arrhythmic Profile in the ΔKPQ Na_V_1.5 Mutant of the Long QT Type 3 Syndrome

**DOI:** 10.3390/biom10060952

**Published:** 2020-06-24

**Authors:** Julio Alvarez-Collazo, Alejandro López-Requena, Julio L. Alvarez, Karel Talavera

**Affiliations:** Laboratory of Ion Channel Research, Department of Cellular and Molecular Medicine, KU Leuven, VIB Center for Brain & Disease Research, 3000 Leuven, Belgium; julioac87@outlook.com (J.A.-C.); alopezr1@yahoo.com (A.L.-R.); julio.alvarez_2007@yahoo.com (J.L.A.)

**Keywords:** hesperetin, flavonoid, citrus, human cardiac Na^+^ channel, Na_V_1.5, genetic variant, mutant, long QT type 3 syndrome, ΔKPQ, precision medicine

## Abstract

Type 3 long QT syndromes (LQT3) are associated with arrhythmogenic gain-of-function mutations in the cardiac voltage-gated Na^+^ channel (hNa_V_1.5). The citrus flavanone hesperetin (HSP) was previously suggested as a template molecule to develop new anti-arrhythmic drugs, as it blocks slowly-inactivating currents carried by the LQT3-associated hNa_V_1.5 channel mutant R1623Q. Here we investigated whether HSP also has potentially beneficial effects on another LQT3 hNa_V_1.5 channel variant, the ΔKPQ, which is associated to lethal ventricular arrhythmias. We used whole-cell patch-clamp to record Na^+^ currents (I_Na_) in HEK293T cells transiently expressing hNa_V_1.5 wild type or ΔKPQ mutant channels. HSP blocked peak I_Na_ and the late I_Na_ carried by ΔKPQ mutant channels with an effective concentration of ≈300 μM. This inhibition was largely voltage-independent and tonic. HSP decreased the rate of inactivation of ΔKPQ channels and, consequently, was relatively weak in reducing the intracellular Na^+^ load in this mutation. We conclude that, although HSP has potential value for the treatment of the R1623Q LQT3 variant, this compound is inadequate to treat the LQT3 associated to the ΔKPQ genetic variant. Our results underscore the precision medicine rationale of better understanding the basic pathophysiological and pharmacological mechanisms to provide phenotype- genotype-directed individualization of treatment.

## 1. Introduction

Sudden cardiac death (SDC) is a major health problem accounting for more than 4,000,000 deaths annually worldwide [1]. In this scenario, arrhythmias of genetic origin, such as long QT syndromes (LQTS), have received special interest due to their lethality. It is known that the lengthening of the QT interval increases the risk of SCD [2], but the prevalence and risk stratification are still incomplete [2,3,4] and the management of LQTS remains challenging. The most common therapeutic approach of LQTS nowadays is pharmacological treatment. The use of β-blockers has been recommended for this purpose, but it is still under debate which of these compounds is best or which patients benefit the most [5,6,7,8,9,10]. A recent meta-analysis suggested a probable genotype-dependent effectiveness of β-blockers [11]. This situation is more complex in type 3 long QT syndromes (LQT3), in which β-blockers are much less effective [12,13,14,15,16]. The fatal arrhythmic events in LQT3 occur at a much higher percentage at low heart rates (during sleep or rest) and antiarrhythmic drugs are currently prescribed together with β-blockers [8,14,17]. However, the use of antiarrhythmic drugs might increase the risk of SCD [18,19], particularly in certain patients (e.g., with a particular LQT3 genotype) [20,21,22].

Precision medicine highlights the importance of understanding the basic pathophysiological and pharmacological mechanisms in the development of better therapeutic approaches based on phenotype- genotype-directed individualization of treatments [23,24]. In this context, LQT3 phenotypes represent an appropriate framework to study the relationship between molecular and clinical events. LQT3 disorders are associated with gain-of-function mutations in the human cardiac voltage-gated Na^+^ channel (hNa_V_1.5). These mutant channels display an increased Na^+^ influx that increases the action potential duration, decreases the repolarization reserve and may give rise to early afterdepolarizations and triggered activity inducing torsades de pointes and ventricular fibrillation [25]. However, the different LQT3 mutations lead to distinct LQT3 phenotypes [26,27]. Channel mutants, such as the R1623Q, show a reduced rate of Na^+^ current inactivation and a small late Na^+^ current (I_Na-Late_) [28,29], while mutants such as the ΔKPQ are characterized by a marked I_Na-Late_ [30]. This raises the possibility that LQT3 patients with different genetic backgrounds (i.e., different mutations) might react differently to the same drug [20,21,22].

In a recent study [29], we showed that hesperetin (HSP), a natural citrus-derived flavonoid, blocked hNa_V_1.5 channels with an effective inhibitory concentration (IC_50_) of 130 μM. HSP also increased the rate of channel inactivation, decreased the net influx of Na^+^ and preferentially inhibited I_Na_ during its slow inactivation phase in a voltage-dependent manner. Remarkably, all these HSP effects were significantly more pronounced in the lethal LQT3-associated mutant R1623Q, suggesting the potential of HSP as a template molecule for the development of novel antiarrhythmic drugs against LQT3 disorders [29]. However, given the distinct nature of LQT3 phenotypes referred above, one cannot assert that HSP would have similar effects on other LQT3 variants, such as the ΔKPQ. The aim of this study was to characterize, with the patch-clamp technique, the effects of HSP on the hNa_V_1.5 channel carrying the ΔKPQ mutation. We found that HSP inhibits hNa_V_1.5 ΔKPQ channels, albeit with lower potency than the wild type (WT) and the R1623Q channels. More importantly, HSP inhibits with similar potency the ΔKPQ peak I_Na_ and I_Na-Late_ and decreases the inactivation rate of this channel. These results demonstrate that HSP has an inadequate antiarrhythmic profile for the treatment of the LQT3 syndrome variant associated with the ΔKPQ mutation.

## 2. Materials and Methods

### 2.1. Cell Culture and Transfection

Human embryonic kidney cells (HEK293T) were seeded on 18 mm glass coverslips previously coated with poly-L-lysine (0.1 mg/L). The cells were grown in a Dulbecco’s modified Eagle’s medium (10% of human serum, 2 U/mL penicillin, 2 mg/mL streptomycin and 2 mM L-glutamine) and stored at 37 °C in a humidity-controlled incubator with 10% CO_2_. Na_V_1.5 mutations were introduced by overlap extension polymerase chain reaction (PCR) and the amplicons containing the mutation were cloned into the pCAGGS-IRES-GFP vector [31] using appropriate restriction sites and sequence-verified. The genes encoding for the human Na_V_1.5 WT or its ∆KPQ mutant were transiently transfected in HEK293T cells using the TransIT-293 reagent (Mirus, Madison, MI, USA). Transfected cells were identified by the GFP expression during the patch-clamp experiments, performed 24–48 h after transfection.

### 2.2. Patch-Clamp Experiments

Whole-cell currents were recorded using only one cell per coverslip. All recordings were carried out at 21–23 °C. We used an EPC7 patch-clamp amplifier (LIST Electronics, Darmstadt, Germany), a TL-1 DMA interface (Axon Instruments) and the pClamp software (Version 9.0, Axon Instruments, Foster City, CA, USA). Currents were filtered with a low-pass at 3 kHz, digitized at 20 kHz and analyzed off-line with the WinASCD software (KU Leuven License, Belgium). The extracellular solutions were applied by gravity as previously described [32]. The control solution contained (in mM): 140 NaCl, 2 CaCl_2_, 1 MgCl_2_, 10 HEPES and 10 glucose, and was titrated to pH 7.4 with NaOH. The intracellular (pipette) solution contained (in mM): 130 CsCl, 1 MgCl_2_, 1 CaCl_2_, 5 EGTA, 5 Na_2_ATP, 5 Na_2_-creatine phosphate, 10 HEPES, and the pH adjusted to 7.2 with CsOH. Patch pipettes were pulled from borosilicate capillary tubes and had resistances in the range 1.5–2 MΩ. The cell membrane capacitance (C_m_ = 22.1 ± 0.9 pF; *n* = 49), and the uncompensated series resistance (R_s_ = 3.5 ± 0.2 MΩ; *n* = 49) were determined using the built-in compensation circuits of the EPC-7 amplifier. R_s_ was electronically compensated up to 50% without ringing and was continually monitored during the experiment. The liquid junction potential was compensated before establishing the gigaseal and the capacity transients were compensated using the pClamp -P/4 protocol. I_Na_ was routinely monitored using 50-ms voltage pulses to −20 mV, applied at 0.25 Hz from a holding potential (HP) of −100 mV. The peak I_Na_ amplitude was measured as the difference between peak inward current and the absolute zero-current level. The late I_Na_ (I_Na-Late_) was measured as the difference between the mean current of ten traces at the end of the 50-ms voltage-clamp pulse and the absolute zero-current level [29]. Current amplitudes of each cell were divided by the membrane capacitance and expressed as current densities (pA/pF).

The percentage of inhibition induced by HSP at different concentrations were determined for the peak and late I_Na_ to obtain the concentration-effect relationships. These data were fit by a Hill function to obtain the effective inhibitory concentration (IC_50_) and the Hill number (H). The time course of the inactivation phase of the I_Na_ traces was fitted by a double exponential function of the form:(1)$$I\left(t\right)=A_{\mathrm{fast}}\cdot e^{-\left(\frac{t}{\tau_{\mathrm{fast}}}\right)}+A_{\mathrm{slow}}\cdot e^{-\left(\frac{t}{\tau_{\mathrm{slow}}}\right)}$$

This procedure yielded two inactivation time constants, τ_fast_ and τ_slow_, and the amplitudes of the two components (A_fast_ and A_slow_) that were used to calculate the relative amplitude of the slow inactivation component [29]. The availability curves and current–voltage relationships (I–V) were obtained by using a standard double-pulse voltage-clamp protocol [29]. The cells were clamped at a HP of −100 mV. The 50-ms test pulse to −20 mV was preceded by 500 ms pre-pulses to various membrane potentials. The availability curves were obtained from the normalization of the amplitude of currents recorded at a given test pulse by the maximal current (I_Max_). These curves were fit by the equation:(2)$$\frac{I\left(V\right)}{I_{\mathrm{Max}}}=\frac{1}{1+\exp\left(\frac{V-V_{\mathrm{inac}}}{s_{\mathrm{inac}}}\right)}$$
where V_inac_ is the voltage for half-maximal availability and s_inac_ is the slope factor.

The I–V relationships were obtained from the amplitude of currents elicited by pre-pulse potentials and were fit by the equation:(3)$$I\left(V\right)=\frac{G_{M}\left(V-V_{\mathrm{Na}}\right)}{1+\exp\left(-\frac{V-V_{\mathrm{act}}}{s_{\mathrm{act}}}\right)}$$
where G_M_ is the maximal whole-cell conductance, V_Na_ is the equilibrium potential for Na^+^, V_act_ is the voltage for half-maximal activation and s_act_ is the slope factor. Activation curves were obtained for each cell by dividing the experimental current amplitudes, I(V), by the corresponding values of G_M_(V − V_Na_) [29].

To investigate tonic and use-dependent blocks of I_Na_ by HSP, cells were voltage-clamped with 50-ms pulses from −100 mV to −20 mV at a frequency of 0.25 Hz. Stimulation was stopped and perfusion with HSP was then initiated. After 1 min, stimulation was resumed (in the presence of HSP) at frequencies of 0.25 or 1 Hz. Tonic block was estimated as the difference between peak control I_Na_ and the I_Na_ at the first pulse after resumption of stimulation during drug perfusion. Use-dependent block was considered to be the difference between peak I_Na_ at the first and the 15th pulse during drug exposure. Tonic and use-dependent blocks are expressed as a percentage of total block, which was estimated from the difference between peak I_Na_ currents recorded in the control and in the presence of HSP.

### 2.3. Data and Statistical Analyses

The data was analyzed and fitted in Origin 9.0 (OriginLab Corporation, Northampton, MA, USA), and was reported as means ± standard errors of means. All data was normally distributed (Shapiro–Wilk test; 95% confidence level). The statistical test and sample numbers are indicated in the text or in the figure legends. Differences were considered statistically significant if *p* < 0.05.

### 2.4. Chemicals

All chemicals were purchased from Sigma-Aldrich. Hesperetin (>98% purity) was kept at 4 °C as a stock solution at 100 mM in dimethyl sulfoxide (DMSO). DMSO had no significant effect on the peak I_Na_ (0.32 ± 0.17% of inhibition) at the maximal concentration used (1%). The hesperetin dilutions in the extracellular solutions were freshly prepared from the stock.

## 3. Results

### 3.1. Characteristics of the ∆KPQ hNa_V_1.5 Currents

The mean density of peak hNa_V_1.5 WT currents recorded at −20 mV in control condition was −87 ± 9 pA/pF (*n* = 12), similar to our previous report [29]. The mean peak I_Na_ density of ∆KPQ hNa_V_1.5 currents recorded in the same conditions was −104 ± 7 pA/pF (*n* = 40). The WT and ∆KPQ mutant currents exhibited similar time constants of inactivation: τ_fast_ = 0.38 ± 0.06 ms and τ_slow_ = 2.9 ± 0.2 ms (*n* = 7) for the WT channel, and τ_fast_ = 0.45 ± 0.03 ms and τ_slow_ = 3.1 ± 0.2 ms for the ∆KPQ mutant (*n* = 12, n.s. vs. WT). However, the relative amplitude of the slow inactivation component was significantly larger for the ∆KPQ 7.0 ± 0.3% (*n* = 12) than for the WT currents 3.1 ± 0.2% (*n* = 7; *p* < 0.05). In addition, the ∆KPQ mutant displayed a more evident I_Na-Late_ (6.9 ± 0.6 pA/pF, *n* = 12) compared to WT (1.3 ± 0.2 pA/pF, *n* = 7, *p* < 0.05; Figure 1a).

### 3.2. Effects of Hesperetin on ∆KPQ hNa_V_1.5 Currents

Extracellular application of 100 µM HSP decreased the amplitude of hNa_V_1.5 WT currents recorded at −20 mV in about 50% (Figure 1b), as previously reported [29]. HSP also decreased the amplitude of currents carried by the ∆KPQ mutant channels, albeit with less potency (Figure 1c). A close inspection of these current traces revealed that HSP increased the rate of inactivation of the WT currents, as previously reported [29], but had the opposite effect on the currents carried by the ∆KPQ mutant. These distinct effects prompted us to investigate the action of HSP on ∆KPQ currents in more detail.

HSP decreased peak WT I_Na_ recorded at −20 mV with an IC_50_ = 116 ± 10 µM (H = 0.8 ± 0.1; *n* = 7), in correspondence to our previous report [29]. HSP also reversibly decreased peak ∆KPQ hNa_V_1.5 currents in a concentration-dependent manner, but with a higher IC_50_ (=320 ± 20 µM; H = 0.6 ± 0.1; Figure 2a,b). The IC_50_ for HSP-induced inhibition of the ∆KPQ I_Na-Late_ was 270 ± 18 µM (H = 0.6 ± 0.1; *n* = 11; Figure 2b). This value was not significantly different from that of peak ∆KPQ I_Na_, but significantly higher than the IC_50_ for the WT I_Na-Late_ (47.9 ± 6.7 µM; H = 0.6 ± 0.1; *n* = 7; similar to our previous report [29]).

Next, we explored the effects of HPS on the time course of inactivation of the ∆KPQ current, using a concentration of 300 µM (a value close to the IC_50_). A comparison of current traces that were recorded in the control and in the presence of HSP and normalized to their respective peak amplitudes shows a clear reduction of the rate of inactivation under the action of the flavonoid (Figure 3a). The fit of the inactivation phase of the currents by a double-exponential function revealed that HSP increased both τ_fast_ and τ_slow_ in a concentration-dependent manner (Figure 3b). This sharply contrasted with the reducing effect of HSP on both time constants of the WT currents, up to 0.39 ± 0.05 ms and 1.7 ± 0.5 ms (*n* = 7) at a concentration of 300 µM, in agreement to our previous findings [29]. We previously reported that HSP reduced the relative amplitude of the slow inactivation component (A_Slow_) of WT currents in a concentration-dependent manner, and that this effect was dramatically enhanced for the currents carried by the R1623Q mutant [29]. In contrast, we found that HPS reduced A_Slow_ of ∆KPQ currents only at the highest concentrations tested (300 and 1000 µM, Figure 3c). To determine the composite effect of the action of HSP on the current amplitude and on the inactivation kinetics on the total Na^+^ influx during the voltage-clamp pulse, we calculated the time integral of the I_Na_ traces. To allow for a proper comparison, we looked at the effects of HSP at concentrations near the IC_50_ for peak I_Na_, i.e., 100 µM for the WT (and the R1623Q mutation) [29] and 300 µM for the ∆KPQ mutant. HSP 100 µM decreased the Na^+^ influx through WT channels by 46.7 ± 1.2% in the WT (*n* = 7), similar to our previous report [29], but only by 18.5 ± 1.8% for ∆KPQ channels (*n* = 21, *p* < 0.05 vs. WT; Figure 3d). At HSP 300 µM the reduction of Na^+^ influx was 69.2 ± 5.5 for the WT (*n* = 7) and 34.3 ± 2.6% for the ∆KPQ mutant (*n* = 12, *p* < 0.05 vs. WT).

To further investigate the action of HSP on the gating properties of ∆KPQ channels, we determined the effects of this compound on the time constants of inactivation at different potentials, on the I–V relationship and on the activation and availability curves. For this we applied a classical double-pulse voltage protocol (see Section 2) in the control and in the presence of 300 µM HSP. In these recordings, application of HSP induced an obvious decrease in the rate of inactivation (Figure 4a). The effects of HSP on the time constant of fast inactivation (τ_fast_) were not voltage-dependent, as the increase induced by HSP did not vary significantly across the tested voltages. However, the increase in the time constant of slow inactivation (τ_slow_) was significantly larger (*p* < 0.05) at more positive potentials (Figure 4b).

HSP (300 µM) significantly reduced the density of ∆KPQ currents in the range of potentials from −40 to +40 mV (*p* < 0.05; one-way ANOVA with Tukey’s post hoc test; *n* = 19; Figure 4a,c). At this concentration HSP reduced the maximal whole-cell conductance (G_M_) of the WT hNa_V_1.5 currents in 75% (*n* = 5) and in 50% the G_M_ of ∆KPQ currents (*n* = 19) (Table 1). HSP did not induce significant changes on the estimated reversal potential (V_Na_) of either channel (Table 1). HSP provoked no changes in the activation and availability curves (Figure 4d) and, correspondingly, the values of V_inac_, s_inac_, V_act_ and s_act_ determined in the control or in the presence of HSP were not significantly different (Table 1). In contrast, and in agreement with our previous report [29], HSP shifted the V_inac_ of the WT channel to more negative potentials (Table 1; *n* = 5).

We found that the inhibitory effect of HSP (300 µM) on ∆KPQ peak and late I_Na_ was poorly dependent on the HP (Figure 5a,b). When the HP was depolarized from −120 mV to −80 mV the inhibition of peak and late I_Na_ by HSP was increased only by ≈29%. In contrast, for the WT currents this increment amounted to 42.3 ± 2.4% for the peak I_Na_ and 34.8 ± 4.4% for I_Na-Late_ (*n* = 5), in agreement with our previous report [29]. The data of the inhibition percentage as a function of the HP were fitted with a Boltzmann function that yielded the voltage for half-maximal effect (V_HP_) and the associated slope factor (s_HP_) [29]. In the WT, V_HP_ and s_HP_ for peak and late I_Na_ were significantly different (−110.4 ± 2.3 mV and −117.2 ± 1.6 mV for V_HP_ and 19.3 ± 2.8 mV and 14.8 ± 1.5 mV for s_HP_ in control and HSP, respectively; *p* < 0.05). In the ∆KPQ both parameters were similar for peak and late I_Na_ (−91.0 ± 1.4 mV and −92.9 ± 3.4 for V_HP_ and 12.1 ± 1.5 mV and 11.2 ± 3.6 mV for s_HP_ in control and HSP, respectively, *n* = 7). V_HP_ values for WT and ΔKPQ were significantly different (*p* < 0.05) in the control and HSP.

Finally, using the protocol to examine use-dependent effects, we found that at stimulation frequencies of 0.25 and 1 Hz, HSP blocked ∆KPQ currents in a mostly tonic fashion. The tonic block amounted to 94.2 ± 1.6% of total block at the lowest frequency (0.25 Hz; *n* = 7) and to 93.5 ± 2.4% of total block at 1 Hz (*n* = 7). In the case of the WT, the tonic block amounted to 93.4 ± 1.2% of total block at 0.25 Hz (*n* = 5) and to 92.7 ± 1.1% of total block at 1 Hz (*n* = 5).

## 4. Discussion

The main finding of the present study is that HSP blocks the human Na_V_1.5 ΔKPQ mutant channel expressed in HEK293T cells, but in a less potent manner than the WT channel and the LQT3 variant R1623Q [29]. Furthermore, and in sharp contrast to the action on the WT and the R1623Q currents, HSP slows down the inactivation of ΔKPQ mutant currents and its effects are not voltage-dependent.

The presence of a marked late component in ΔKPQ currents [27,30] or a decreased inactivation rate in R1623Q currents [27,28] constitutes the basis of the larger action potential duration, characteristic of the LQT3 syndrome [26,33]. High rate arrhythmias in LQT3 syndrome (e.g., torsades de pointes) occur as a consequence of the increased action potential duration that favours the appearance of early afterdepolarizations and, ultimately, lethal arrhythmic events.

We showed previously that HSP inhibits I_Na_ in the LQT3 mutant R1623Q and increased the rate of inactivation at low concentrations [29]. Consequently, Na^+^ load was strongly reduced by HSP in cells expressing this mutation. Moreover, the HSP effect was voltage-dependent [29]. These actions on the R1623Q channel variant make HSP an interesting molecular template for the treatment of lethal arrhythmias typical of the LQT3 syndromes. However, the present results obtained on the ΔKPQ variant differ considerably. First, the IC_50_ for peak I_Na_ inhibition was higher for the ΔKPQ mutant than for HSP on the WT and the R1623Q channels (320 vs. 130 µM) and for the commonly used antiarrhythmic drugs (lidocaine, disopyramide [34] and flecainide [35]) on the WT channel. Second, the HSP inhibition potencies on I_Na-Late_ and peak I_Na_ where similar for the ΔKPQ channels, whereas this compound was more potent on the I_Na-Late_ of the WT and R1623Q channels (IC_50_ ≈ 50 µM and 30 µM, respectively) [29]. The preferential inhibition of the I_Na-Late_ is a recognized advantageous feature for class I antiarrhythmic drugs (hNa_V_1.5 blockers; e.g., ranolazine) [36,37,38]. Third, HSP effects were not voltage-dependent in ΔKPQ channels: availability and activation curves were not shifted by HSP and decreasing the HP to −80 mV barely increased peak I_Na_ or I_Na-Late_ inhibition. A voltage-dependent action could be important in controlling action potential duration in several pathological states but particularly in LQT3 syndromes. Fourth, and most important, contrary to what was shown for WT and R1623Q channels [29], HSP decreased the rate of inactivation of ΔKPQ. It must be noted here that the increase in the inactivation rate of the R1623Q mutant channel [29] also decreases the intracellular Na^+^ overload and therefore the use of HSP could be a good therapeutic strategy for the long-term treatment of LQT3 syndrome [39]. In this sense, it is obvious that the action of HSP on the ΔKPQ phenotype is not therapeutically attractive, since the decrease in net Na^+^ influx in this variant was very discreet (≈18%), compared to WT (≈47%) and to R1623Q (≈65%) [29]. The HSP-induced decrease of the inactivation rate implies a higher Na^+^ influx, resulting in the modest capacity of HSP to reduce the Na^+^ load by activation of the ΔKPQ channel variant.

Our results indicate that HSP action is dependent on the channel variant and highlights the relevance of understanding the basic pathophysiological and pharmacological mechanisms for the design of adequate and specific therapeutic approaches for each particular LQT3 syndrome genotype. However, our results do not allow for the advancing of an explanation of how HSP may have different effects on the WT, the single residue mutation R1623Q and the multiple deletion ΔKPQ channels. Based on the results of patch-clamp experiments and on molecular docking simulations, we previously suggested that HSP has access from the membrane to the LA binding site (residue F1760) [29] via the lateral fenestrations of the Na_V_1.5 pore [40,41,42]. Determining whether this blockade pathway could give rise to distinct effects in different mutations (and different channel conformations) requires further investigation.

## 5. Conclusions

In our previous study on the R1623Q LQT3 mutation [29] we proposed HSP as a template for the development of antiarrhythmic drugs against lethal arrhythmias in LQT3. However, the present results indicate that HSP possesses an inadequate antiarrhythmic profile to treat the ΔKPQ variant of the LQT3 syndrome. These results should be interpreted as support for the idea of a phenotype- genotype-directed individualization of treatment based on a better understanding of the basic pathophysiological and pharmacological mechanisms.

## Figures and Tables

**Figure 1 biomolecules-10-00952-f001:**
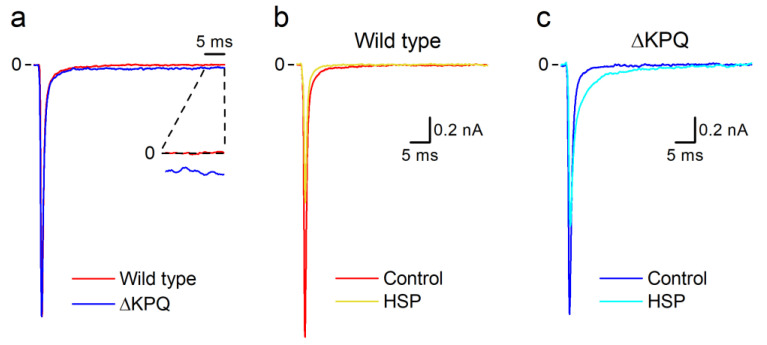
Comparison of human WT and LQT3 Na_V_1.5 currents. (**a**) Representative I_Na_ traces recorded in HEK293T cells expressing WT hNa_V_1.5 channels or the hNa_V_1.5 LQT3 variant ∆KPQ. Each trace was normalized with respect to peak inward current. The inset shows the last 5-ms interval of the current traces amplified (10×). Note that the ∆KPQ LQT3 variant presents a larger amplitude of I_Na-Late_ compared to WT. (**b**,**c**) Comparison of the effects of 100 µM HSP on WT (**b**) and ΔKPQ (**c**) Na_V_1.5 currents. Currents were evoked with 50-ms voltage-clamp pulses to −20 mV from a holding potential of −100 mV. Pulse rate 0.25 Hz.

**Figure 2 biomolecules-10-00952-f002:**
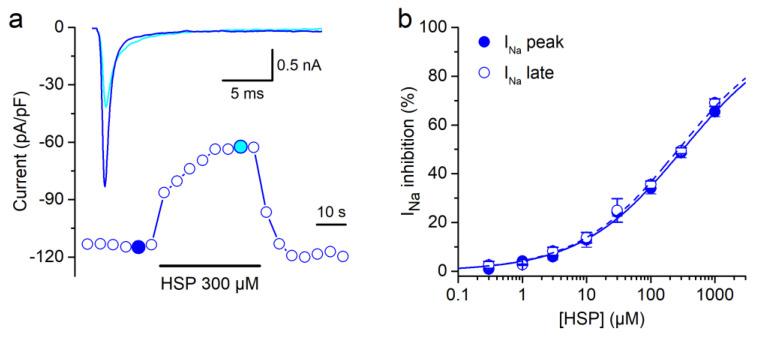
Hesperetin inhibits currents carried by the LQT3 ∆KPQ mutant channels. (**a**) Time course of the I_Na_ recorded in HEK293T cells expressing ∆KPQ hNa_V_1.5 in the control condition and in the presence of HSP 300 µM. Note that the effect of HSP was fully reverted upon washout. Representative current traces in control and upon the application of HSP are shown in the inset and correspond with the colored data points. For clarity only the first 25 ms of the current traces are shown. Note that HSP decreased peak I_Na_ but also slowed down its inactivation. (**b**) Concentration–effect curves for the action of HSP on ∆KPQ hNa_V_1.5 peak and late currents. The dots represent the mean ± s.e.m. inhibition percentage during the application of different concentrations of HSP (*n* = 21). The line represents the Hill fit of the data. Cells were clamped at a holding potential of −100 mV and 50-ms pulses to −20 mV were continuously applied at a rate of 0.25 Hz.

**Figure 3 biomolecules-10-00952-f003:**
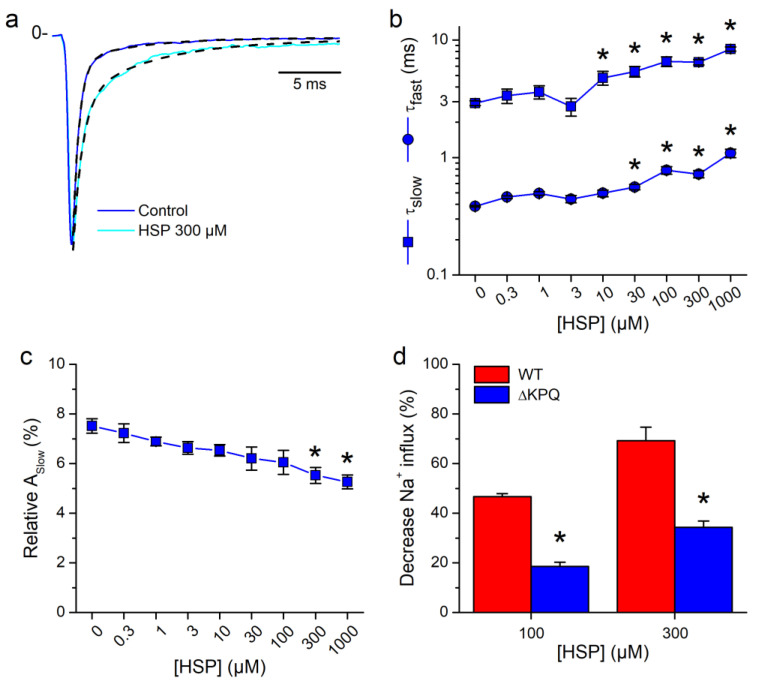
Hesperetin decreases the inactivation rate of LQT3 ∆KPQ mutant currents. (**a**) Superimposed ∆KPQ hNa_V_1.5 currents in the control and in the presence of HSP 300 µM. Note that HSP decreases inactivation rates. The dotted lines represent bi-exponential fittings of current traces. (**b**) Effects of different concentrations of HSP on inactivation time constants of I_Na_ in the ∆KPQ hNa_V_1.5 channels. * *p* < 0.05 compared to its own control; *n* = 21, one-way ANOVA with Tukey’s post hoc test. (**c**) HSP barely decreased the amplitude of the slow component of I_Na_ inactivation (A_slow_). A significant (*p* < 0.05) reduction was only achieved at the highest concentrations. (**d**) The decrease in Na^+^ influx by HSP is more marked in WT hNa_V_1.5 channels. Columns are the mean (±s.e.m.) percent decreases in Na^+^ influx by HSP with respect to control values. HSP discreetly reduced the amount of Na^+^ influx during the flow of I_Na_, in hNa_V_1.5 channels carrying the ∆KPQ mutation.

**Figure 4 biomolecules-10-00952-f004:**
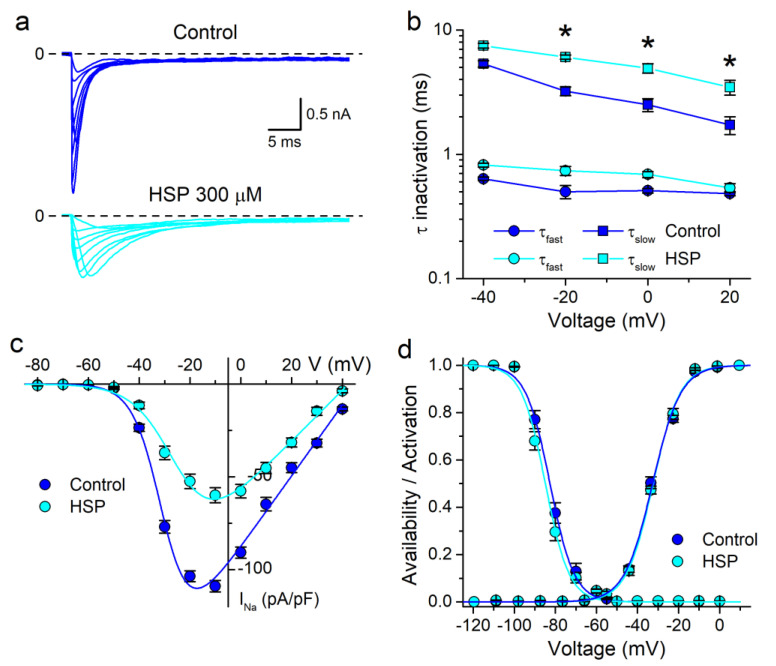
Lack of voltage-dependent effects of hesperetin on I_Na_ in the ∆KPQ mutant channels. (**a**) Examples of ∆KPQ hNa_V_1.5 currents elicited by voltage pulses applied from a holding potential of −100 mV to between −40 and +30 mV in 10 mV steps, in the control and in the presence of HSP 300 µM. (**b**) Voltage-dependency of τ_fast_ and τ_slow_ in control and in the presence of HSP (300 µM). The action of HSP on τ_fast_ was not voltage-dependent. The increase in τ_slow_ by HSP was more marked at potentials between −20 and 20 mV (* *p* < 0.05 with respect to the effect at −40 mV). (**c**) I–V relationships from ∆KPQ hNa_V_1.5 expressing HEK293T cells in the control condition and in the presence of HSP 300 µM. The dots represent the mean ± s.e.m. (*n* = 19). HSP significantly decreased the current in the voltage range from −40 to +40 mV. (**d**) Availability and activation curves for ∆KPQ hNa_V_1.5 channel in the control and in the presence of HSP 300 µM. The dots represent the mean ± s.e.m. (*n* = 19).

**Figure 5 biomolecules-10-00952-f005:**
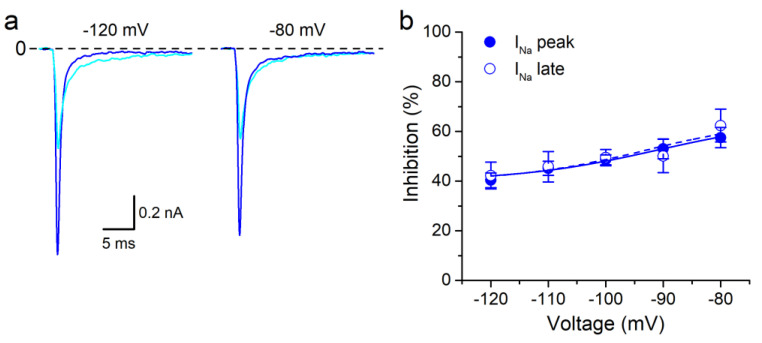
The effect of HSP on I_Na_ in the ∆KPQ mutant channels is poorly dependent on the holding potential. (**a**) Effects of HSP (300 µM) on I_Na_ evoked at −20 mV from two different holding potentials (HP). Decreasing the HP from −120 mV to −80 mV, barely increased the block of I_Na_ by HSP. (**b**) Boltzmann functions describing the HP dependency of HSP (300 µM) effect on peak I_Na_ and late I_Na_ in ∆KPQ hNa_V_1.5 channel. The dots represent the mean ± s.e.m. (*n* = 7). The inhibitory effect of HSP is barely dependent on the holding potential in ΔKPQ channels.

**Table 1 biomolecules-10-00952-t001:** Parameters obtained from the fits of the I–V relationships and availability curves.

Condition	G_M_ (nS/pF)	V_Na_ (mV)	V_act_ (mV)	s_act_ (mV)	V_inac_ (mV)	s_inac_ (mV)
WT control	2.0 ± 0.6	42.6 ± 1.2	−35.8 ± 1.1	5.5 ± 0.7	−70.2 ± 0.4	6.6 ± 0.4
WT HSP	0.5 ± 0.1 ^1^	39.3 ± 2.9	−30.2 ± 0.6	5.6 ± 0.5	−86.0 ± 0.5 ^1^	5.1 ± 0.5
∆KPQ control	1.6 ± 0.3	44.8 ± 1.0	−29.3 ± 0.4	6.4 ± 0.3	−82.6 ± 0.3	5.9 ± 0.3
∆KPQ HSP	0.8 ± 0.1 ^1^	42.3 ± 0.6	−29.1 ± 0.3	6.1 ± 0.2	−84.9 ± 0.5	5.8 ± 0.4

^1^*p* < 0.05 with respect to control; paired *t*-test. WT, *n* = 5; ∆KPQ, *n* = 19.

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
