# Peer review of "The Citrus Flavonoid Hesperetin Has an Inadequate Anti-Arrhythmic Profile in the ΔKPQ NaV1.5 Mutant of the Long QT Type 3 Syndrome"

_biomolecules, 2020, doi:10.3390/biom10060952_

Round 1

Reviewer 1 Report

Dear Authors,

The manuscript on citrus flavonoid hesperetin is well written, and it represents a quality investigation.

I found nothing disturbing in this manuscript, no typing errors, and very good level of language.

I recommend this manuscript to be accepted for publication in its present form.

Author Response

We thank the Reviewer for the positive appreciation of our manuscript.

Reviewer 2 Report

As a practicing clinician, I view the information presented in this article as relevant and scientifically sound. I also applaud the authors for publishing their negative results. As far as I know, the used methods are sound; however, I recommend further review by a basic scientist in the field of arrhythmia. 

Author Response

(The authors gave the same response as above.)

Reviewer 3 Report

New pharmacological treatments for arrhythmic disorders like the long QT syndrome are necessary for beneficial treatment in future. Here, the authors tested the natural flavonoid hesperetin that was a promising candidate for treatment of the LQT3-associated mutant R1623Q in a different model with a Nav1.5 channel carrying the ΔKPQ mutation. Using the patch-clamp technique they clearly demonstrated that hesperetin is inadequate to treat the LQT3-associated mutant ΔKPQ. They appropriately concluded that a genotype-directed treatment may be a possible future.

I think the hypothesis that hesperetin may be a general base for development of new anti-arrhythmic drugs has been clearly refuted. Therefore, I have only one minor point.

The sentence from line 290-292 (…was higher for the ΔKPQ mutant than for HSP…) seems not correct and has to be rewritten.

Author Response

We thank the Reviewer for the positive appreciation of our manuscript.

We corrected the wrong sentence to: "First, the IC50 for peak INa inhibition was higher for the ΔKPQ mutant than for HSP on the WT and the R1623Q channels (320 vs 130 µM) and for the commonly used antiarrhythmic drugs (lidocaine, disopyramide [34] and flecainide [35]) on the WT channel."